# PDE5 inhibitor potentially improves polyuria and bladder storage and voiding dysfunctions in type 2 diabetic rats

**Takafumi Kabuto, So Inamura, Hisato Kobayashi, Xinmin Zha, Keiko Nagase, Minekatsu Taga, Masaya Seki, Nobuki Tanaka, Yoshinaga Okumura, Osamu Yokoyama, Naoki Terada**[ID]*

Department of Urology, Faculty of Medical Science, University of Fukui, Fukui, Japan

* nterada@u-fukui.ac.jp

**Data Availability Statement:** Data are available from the University of Fukui Institutional Data Access. Data contain potentially identifying or sensitive patient information. A Research Ethics

## Abstract

### Purpose

Bladder dysfunction associated with type 2 diabetes mellitus (T2DM) includes urine storage and voiding disorders. We examined pathological conditions of the bladder wall in a rat T2DM model and evaluated the effects of the phosphodiesterase-5 (PDE-5) inhibitor tadalafil.

### Materials and methods

Male Otsuka Long-Evans Tokushima Fatty (OLETF) rats and Long-Evans Tokushima Otsuka (LETO) rats were used as the T2DM and control groups, respectively. Tadalafil was orally administered for 12 weeks. Micturition behavior was monitored using metabolic cages, and bladder function was evaluated by cystometry. Bladder blood flow was evaluated by laser speckle imaging, and an organ bath bladder distention test was used to measure adenosine triphosphate (ATP) release from the bladder urothelium. The expression levels of vesicular nucleotide transporter (VNUT), hypoxia markers, pro-inflammatory cytokines and growth factors in the bladder wall were measured using real-time polymerase chain reaction and enzyme-linked immunosorbent assay. Bladder wall contractions in response to KCl and carbachol were monitored using bladder-strip tests.

### Results

With aging, OLETF rats had higher micturition frequency and greater urine volume than LETO rats. Although bladder capacity was not significantly different, non-voiding bladder contraction occurred more frequently in OLETF rats than in LETO rats. Bladder blood flow was decreased and ATP release was increased with higher VNUT expression in OLETF rats than in LETO rats. These effects were suppressed by tadalafil administration, with accompanying decreased HIF-1α, 8-OHdG, IL-6, TNF-α, IGF-1, and bFGF expression. The impaired contractile responses of bladder strips to KCl and carbachol in OLETF rats with aging were restored by tadalafil administration.

Committee in University of Fukui has imposed them. E-mail address: rinsho-rinri@ml.u-fukui.ac.jp.

**Funding:** The study was conducted with financial support from Nippon Shinyaku. The funders had no role in study design, data collection and analysis, decision to publish, or preparation of the manuscript.

**Competing interests:** NO authors have competing interests.

## Conclusions

The T2DM rats had polyuria, increased ATP release induced by decreased bladder blood flow and impaired contractile function. PDE5 inhibition improved these changes and may prevent T2DM-associated urinary frequency and bladder storage and voiding dysfunctions.

## Introduction

Many epidemiological studies have shown that non-urological disorders such as hypertension, type 2 diabetes mellitus (T2DM), and dyslipidemia are associated with lower urinary tract symptoms in both men and women [1]. Lifestyle factors, and especially the presence of T2DM, can affect lower urinary tract function. In T2DM, lower urinary tract dysfunction has been evaluated using urodynamic studies, revealing detrusor overactivity (DO) in 55%–61% of individuals with T2DM, detrusor underactivity (DU) in 9%–23%, and areflexia in 9%–23% [2, 3]. Moreover, DO and DU are mixed in females with T2DM [4]. According to a review by Daneshgari et al., a mixture of lower urinary tract disorders can be present in T2DM, often with storage disorders in the early stages and voiding disorders in the later stages [5].

Clinical and basic studies suggest that atherosclerosis in both men and women induces decreased bladder blood flow, leading to chronic ischemia of the bladder [6]. Chronic bladder ischemia causes oxidative stress, resulting in bladder denervation and the two in the bladder wall, and leading to DO that progresses to DU. However, the mechanisms underlying how chronic bladder ischemia causes the development of DO have not yet been elucidated. In our earlier study of salt-sensitive hypertensive rats, hypertensive-related bladder ischemia resulted in a decrease in bladder capacity and an increased release of adenosine triphosphate (ATP) from the bladder urothelium [7]. We thus ask the question: does the same mechanism underlie the development of DO in other T2DM models? Most of the research on animal models of T2DM has been conducted using rabbits and rats in which arteriosclerosis was artificially created; few basic studies using T2DM models have been published. We have used Otsuka Long-Evans Tokushima Fatty (OLETF) rats as a pathological model of T2DM and Long-Evans Tokushima Otsuka (LETO) rats as a control. The OLETF rat model develops insulin resistance by 12 weeks of age, hyperinsulinemia by 25 weeks of age, and subsequently has decreased insulin levels [8]. Hyperglycemia is maintained throughout the disease course. The plasma insulin and glucose levels of 36- and 48-week-old OLETF and LETO rats are described in our previous report [9]. The features of this pathological model closely resemble the natural history of human T2DM.

Using this rat model of T2DM, we have previously reported that T2DM-induced chronic ischemia leads to oxidative stress, resulting in prostate enlargement through the upregulation of several cytokines. Treatment with the phosphodiesterase-5 (PDE5) inhibitor tadalafil improves prostate ischemia and might prevent its enlargement via the suppression of cytokines and growth factors [10]. Bladder ischemia-induced chronic inflammation may also be associated with the development of DO and DU [11]. In the present study, we therefore aimed to elucidate (*i*) the mechanisms of DO and DU development in T2DM, and (*ii*) whether DO and DU can be improved by long-term treatment with tadalafil.

The bladder urothelium has an important role in mechanosensory transduction [12, 13]. In response to mechanical stimuli, ATP is released from epithelial cells and activates purinergic receptors on submucosal afferent fibers, thus facilitating bladder sensory signaling [14]. It has been reported that ATP release is mainly regulated by vesicular nucleotide transporter

(VNUT) in the bladder urothelium [15]. Furthermore, although bladder ischemia results in an increase in pro-inflammatory cytokines and growth factors, it remains unknown how this phenomenon is involved in bladder function. In the current study, we therefore evaluated bladder blood flow; ATP release from the bladder urothelium; bladder expression levels of VNUT, hypoxia markers, and pro-inflammatory cytokines and growth fators; and bladder contraction using the T2DM rat model with tadalafil treatment.

## Materials and methods

### Animal preparation

Four-week-old male OLETF and LETO rats were purchased from SLC Inc. (Shizuoka, Japan). LETO rats are a control strain that do not have characteristics of T2DM. The animals were housed at the University of Fukui Animal Center at a constant temperature of 23˚C and 50%–60% humidity with a normal 12-hr light/dark schedule. Tap water and standard rat chow were freely ingested. All animal experiments were conducted in accordance with the guidelines established by the Fukui University Committee for Animal Experimentation (Permission no: R01056).

Oral tadalafil (100 μg/kg/day) was administered to the OLETF and LETO rats for 12 consecutive weeks, beginning when the rats were 36 weeks old; the sham group received only the vehicle for 12 weeks. The experimental dose (100 μg/kg/day) was set based on the standard treatment dose of tadalafil in Japanese patients (5 mg/day) and the average body weight in Japan (60 kg), leading to a calculated standard dose of 83 μg/kg.

In total, 54 rats were divided into the following nine groups (6 rats/group), as depicted in Fig 1A: one group each of L-36 (36-week-old LETO rats), L-48 (48-week-old LETO rats treated with vehicle for 12 weeks), and L-48(t) (48-week-old LETO rats treated with tadalafil for 12 weeks), and two groups each of O-36 (36-week-old OLETF rats), O-48 (48-week-old OLETF rats treated with vehicle for 12 weeks), and O-48(t) (48-week-old OLETF rats treated with tadalafil for 12 weeks).

### Micturition behavior

A metabolic cage was used to measure the voiding parameters of the rats. Rat urine was collected through a urine collection funnel and weighed on an electronic balance. The cumulative weight of the collected urine was recorded every 10 min. The rats were kept in the metabolic cages for approximately 60 hr to acclimate to the cage, and the values recorded in the last 24 hr of that period were used. Each monitoring period started at 18:00. The data collected were used to calculate the mean voided volume and the number of micturitions per 24 hr. Residual urine volume was examined by ultrasonography.

### Cystometry in conscious rats

Cystometry procedures in conscious rats were performed as our previous reports [16]. Prior to cystometry, a catheter was surgically implanted into the bladder of each rat. The catheter was made of polyethylene tubing and was inserted via a lower abdominal incision. After surgery, sufficient time was provided for the rat to recover from anesthesia. Next, he bladder catheter was connected to a pressure transducer via a T-tube. The pressure transducer was used to convert changes in bladder pressure into electrical signals that were recorded. Cystometry was done with physiological saline at room temperature at 0.04 ml per minute. Bladder capacity and bladder contraction pressure were measured.

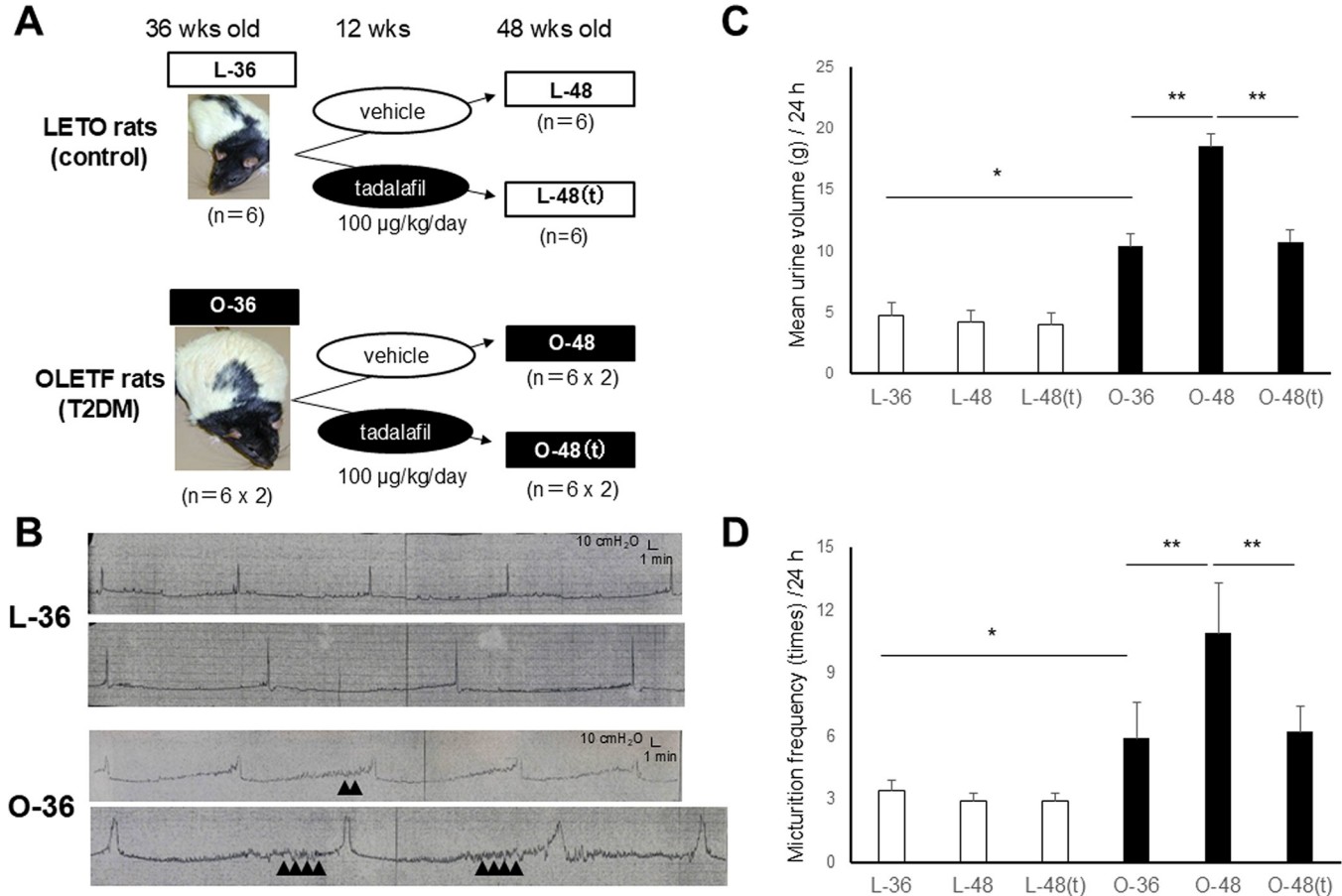

**Fig 1. Measurement of micturition behavior in the rats. A:** The schematic study design of OLETF and LETO rats treated with vehicle or tadalafil. L-36: 36-week-old LETO rats, L-48: 48-week-old LETO rats treated with vehicle for 12 weeks, L-48(t): 48-week-old LETO rats treated with tadalafil for 12 weeks, O-36: 36-week-old OLETF rats, O-48: 48-week-old OLETF rats treated with vehicle for 12 weeks, O-48(t): 48-week-old OLETF rats treated with tadalafil for 12 weeks. Six rats in each LETO rat group and 12 (6 × 2) rats in each OLETF rat group were included. **B:** Cystometry in conscious LETO and OLETF rats at 36 weeks. Non-voiding contractions are indicated as arrows. **C:** Total urine volume for 24 hr in the LETO and OLETF rats. **D:** Micturition frequency for 24 hr in the LETO and OLETF rats. Data are shown as the mean ± SEM. *p<0.05, **p<0.01.

### Laser speckle blood flow imaging system

The blood flow of the bladders of LETO and OLETF rats was evaluated using a laser speckle blood flow imaging system, the Omegazone™ (OZ-2, Omegawave Inc., Tokyo, Japan), as described by Forrester et al. [17]. Each rat was anesthetized using halothane. The bladder was exposed and a catheter was then inserted through the bladder dome. The catheter was connected to the infusion pump, and saline was infused into the bladder until the pressure rose to 10 cm $H_2O$. The bladder surface was diffusely irradiated with a 780-nm semiconductor laser, and the scattered light was treated with a hybrid filter and detected using a charge-coupled device camera. A single blood flow image was generated by averaging the numbers obtained from 20 consecutive raw speckle images. The total blood flow of the entire bladder was then calculated by summing the values on one side and those on the other side of the surface of the bladder.

### Organ bath bladder distention test

After rats were sacrificed by decapitation at 48 weeks of age, the bladder and urethra were removed and weighed. The amount of ATP released from the stretched bladder urothelium

was then measured according to the method of Tanaka et al. [18] with slight modifications. Briefly, one end of the infusion catheter was inserted through the urethra into the bladder, and the bladder neck was secured with surgical sutures. The bladder urothelium was rinsed three times with 0.3 mL of Krebs solution before the catheter was connected to the infusion pump and pressure transducer. The bladder was then fixed vertically in a 10-mL organ bath in which 0.3 mL of Krebs solution was gasified with 5% $CO_2$ and 95% $O_2$ at 37˚C. Next, 1.5 mL of Krebs solution was infused into the bladder at a rate of 0.04 mL/s. The solution in the bladder was then collected on ice using gravity. ATP was measured using the ATPlite™ luciferin-luciferase assay with a Fusion luminometer (Perkin Elmer, Waltham, MA, USA) according to the manufacturer's instructions. The amount of ATP released was converted to the concentration per mg of tissue.

## Real-time polymerase chain reaction (PCR)

The bladders of OLETF rats (O-36, O-48, and O-48(t)) were used to measure mRNA expression. After the rats were sacrificed by decapitation, bladder tissue was cut into small pieces and ground into powder using a mortar and pestle under liquid nitrogen. Total RNA was isolated from the tissue using the RNeasy Fibrous Tissue Mini Kit (Qiagen, Hilden, Germany) according to the manufacturer's instructions. For single-strand complementary DNA synthesis, 2 μg of total RNA was used with the High Capacity RNA-to-cDNA Kit (Applied Biosystems, Foster City, CA, USA) according to the manufacturer's protocol, in a final volume of 20 μL. The reverse transcription step was at 37˚C for 60 min, followed by an inactivation step at 95˚C for 5 min. The mRNA expression was quantitatively analyzed using the SYBR green fluorescence method with an ABI 7300 Real-Time PCR System (Applied Biosystems), with glyceraldehyde 3-phosphate dehydrogenase as an internal control. Differences in the mRNA expression of VNUT, hypoxia-inducible factor-1 alpha (HIF-1α), interleukin-6 (IL-6), tumor necrosis factor alpha (TNF-α), insulin-like growth factor-1 (IGF-1), and basic fibroblast growth factor (bFGF) in the bladder were determined. Each fold change was calculated and used for the statistical analysis. The forward and reverse primers are listed in S1 Table.

## Enzyme-linked immunosorbent assay (ELISA)

Bladder tissue (15 mg) was cut into small pieces and ground into powder using a mortar and pestle under liquid nitrogen. The soluble and insoluble protein fractions were extracted using the CelLytic™ MEM Protein Extraction Kit (Sigma-Aldrich Japan, Tokyo, Japan) and the CelLytic™ NuCLEAR™ Extraction Kit (Sigma-Aldrich Japan), respectively, according to the manufacturer's instructions. IL-6, TNF-α, IGF-1, and bFGF were measured using ELISA with the Rat IGF-1 ELISA Kit (ab213902, Abcam, Cambridge, UK), Rat IL-6 ELISA Kit (ab100772, Abcam), Rat TNF alpha ELISA Kit (ab100785, Abcam), Rat bFGF ELISA Kit (Invitrogen, Waltham, MA, USA), and AssayMax™ Dihydrotestosterone ELISA Kit (Assaypro, St. Charles, MO, USA), respectively. Protein concentrations were calculated as picograms of protein per milligram of tissue wet weight.

## 8-hydroxy-2'-deoxyguanosine (8-OHdG) measurement

For the measurement of 8-OHdG, bladder tissue was homogenized with 0.1 M phosphate buffer containing 1 mM ethylenediaminetetraacetic acid. After centrifugation for 10 min, the supernatant was purified using DNeasy Blood & Tissue Kits (Qiagen) according to the manufacturer's instructions; this was followed by DNA digestion using Nuclease P1 (Wako, Tokyo, Japan). After the addition of 1 unit of alkaline phosphatase (Wako) per 100 μg of DNA and

incubation at 37˚C for 30 min, the samples were boiled for 10 min and kept on ice until use. The levels of 8-OHdG were determined using an 8-OHdG ELISA kit (ab201734, Abcam).

## In vitro bladder-strip experiments

For the in vitro bladder-strip experiments, OLETF rats (i.e., O-36, O-48, and O-48(t) rats) were used; these were different from the rats used for the investigation of ATP release from the bladder urothelium. After the rats were sacrificed by decapitation, full-thickness longitudinal bladder strips (7 mm long × 3 mm wide) were excised and mounted in a 10-mL organ bath containing Krebs solution at 37˚C and continuously bubbled with 95% $O_2$ and 5% $CO_2$. The strips were equilibrated under a resting tension of 1 g for 45 min, and the Krebs solution was changed every 15 min. Following equilibration, the strips were exposed to a solution containing high $K^+$ (62 mM KCl) to normalize the contractile force. After washout of the KCl solution, the isometric contractions of the strips caused by the cumulative application of carbachol ($3 \times 10^{-7}$, $10^{-6}$, $3 \times 10^{-6}$, $10^{-5}$, and $3 \times 10^{-5}$ M) were recorded via force transducers (Nihon Kohden, Tokyo, Japan) and Labchart & Scope software (ADInstruments, Colorado Springs, CO, USA). After the experiment, the bladder strips were weighed to normalize the contractile force.

## Data analysis

The results are presented as the mean ± standard error of the mean (SEM). All data were analyzed with analysis of variance (ANOVA) and independent t-tests using SPSS for Windows, Version 16.0 (SPSS Inc., Chicago, IL, USA). For all analyses, $p < 0.05$ was considered significant.

## Results

### Body weights, bladder wet weights, and micturition characteristics of LETO and OLETF rats

The respective body weights of the LETO and OLETF rats (n = 6, each) were 546 ± 28 g and 671 ± 31 g ($p < 0.01$) at 36 weeks, and 557 ± 27 g and 658 ± 32 g ($p < 0.01$) at 48 weeks; they were significantly higher in the OLETF rats. The respective bladder wet weights of the LETO and OLETF rats were 86 ± 3 mg and 175 ± 7 mg ($p < 0.01$) at 36 weeks, and 103 ± 5 mg and 196 ± 14 mg at 48 weeks ($p < 0.01$); they were significantly higher in the OLETF rats. In both groups, tadalafil treatment did not change the body weight or bladder wet weight of the rats. The mean voided urine volume per body weight was not significantly different between the LETO and OLETF rats (2.7 ± 0.2 mL and 2.8 ± 0.6 mL, respectively; p = 0.678) and did not change with age or tadalafil treatment.

Cystometry in conscious rats was performed in the LETO and OLETF rats (n = 2, each) at 36 weeks. The bladder capacity (0.7 ± 0.1 mL and 0.7 ± 0.2 mL, p = 0.37), and the bladder contraction pressure (39.3 ± 10.7 cmH$_2$O and 33.4 ± 10.3 cmH$_2$O, p = 0.16) were not significantly different between LETO and OLETF rats. Non-voiding contractions appeared only in OLETF rats and not appeared in LETO rats (Fig 1B). There was no residual urine in OLETF or LETO rats during the experiments.

The total urine volume per 24 hr was significantly higher in the OLETF rats than in the LETO rats (n = 6, each) at both 36 (10.4 ± 1.8 mL and 4.8 ± 2.3 mL, $p < 0.01$) and 48 (18.5 ± 3.8 mL and 4.2 ± 0.7 mL, $p < 0.01$) weeks (Fig 1C). The frequency of micturition per 24 hr was also significantly higher in the OLETF rats than in the LETO rats at both 36 (5.9 ± 1.7 and 3.3 ± 1.6, $p < 0.01$) and 48 (10.9 ± 2.4 and 2.9 ± 0.4, $p < 0.01$) weeks (Fig 1D).

The total urine volume and frequency of micturition in the OLETF rats were significantly higher at 48 weeks than at 36 weeks (p<0.01). Furthermore, tadalafil treatment significantly decreased the total urine volume (from 18.5 ± 3.8 mL to 10.7 ± 2.0 mL, p<0.05) and frequency of micturition (from 10.9 ± 2.4 to 6.0 ± 1.2, p<0.05) at 48 weeks in the OLETF rats. By contrast, the total urine volume and frequency of micturition in the LETO rats were not significantly different between 36 and 48 weeks, and were unchanged by tadalafil treatment (Fig 1C, 1D).

## ATP production and VNUT expression in the bladder

To clarify the mechanisms underlying the differences in bladder contraction between OLETF and LETO rats, we measured the ATP concentrations released from the bladder urothelium using the organ bath bladder distention test (n = 3, each) (Fig 2A). The ATP concentrations of the OLETF rats were significantly higher than those of the LETO rats (p<0.01), and ATP concentrations were significantly decreased by tadalafil treatment in both LETO (p<0.05) and OLETF (p<0.01) rats (Fig 2B). These results indicate that ATP production from the bladder urothelium is associated with frequent bladder contraction in OLETF rats, and is improved by tadalafil treatment.

To clarify the mechanisms associated with the changes in ATP production, we evaluated the bladder expression of VNUT. At 36 weeks, VNUT expression was significantly higher in the OLETF rats than in the LETO rats (p<0.05) (Fig 2B); at 48 weeks, VNUT expression was significantly increased in both LETO (p<0.05) and OLETF (p<0.01) rats. Moreover, VNUT expression was significantly decreased by tadalafil treatment in both LETO (p<0.01) and OLETF (p<0.01) rats. These results suggest that changes in bladder VNUT expression might be associated—at least in part—with ATP production in the bladder urothelium.

## Changes in bladder blood flow measured by laser speckle blood flow imaging

Blood flow to the bladder was measured using the Omegazone laser speckle blood flow imaging system just before the bladder was extracted from each rat (n = 3, each) (Fig 3A). The blood flow is displayed as changes in laser frequency in the speckle imaging. The Omegazone uses pixels of different colors: blue indicates areas with poor blood flow and yellow indicates areas with high blood flow (Fig 3B). The imaging results revealed that blood flow was significantly lower at 48 weeks than at 36 weeks in both LETO (p<0.01) and OLETF (p<0.005) rats. At 48 weeks, blood flow was significantly lower in the OLETF rats than in the LETO rats (p<0.01). Tadalafil treatment produced significant increases in blood flow to the bladder in both the LETO (p<0.01) and OLETF (p<0.05) rats (Fig 3C). These results indicate that, in older rats, blood flow to the bladder is lower in OLETF rats than in LETO rats, and is increased by tadalafil treatment.

## Expression levels of HIF-1α and 8-OHdG in the bladders of OLETF rats

To evaluate bladder hypoxia induced by changes in blood flow, we evaluated HIF-1α expression levels in the bladders of OLETF rats (n = 3, each); HIF-1α expression was significantly higher at 48 weeks than at 36 weeks (p<0.01). Although HIF-1α expression tended to be decreased by tadalafil treatment, this difference was not significant (Fig 4A). 8-OHdG is an oxidative stress marker that is made when DNA is damaged by reactive oxygen species. In the bladders of OLETF rats, 8-OHdG levels were significantly higher at 48 weeks than at 36 weeks (p<0.01) and were significantly decreased by tadalafil treatment (p<0.01) (Fig 4B). These results indicate that hypoxia is induced by decreased blood flow in the bladders of older OLETF rats.

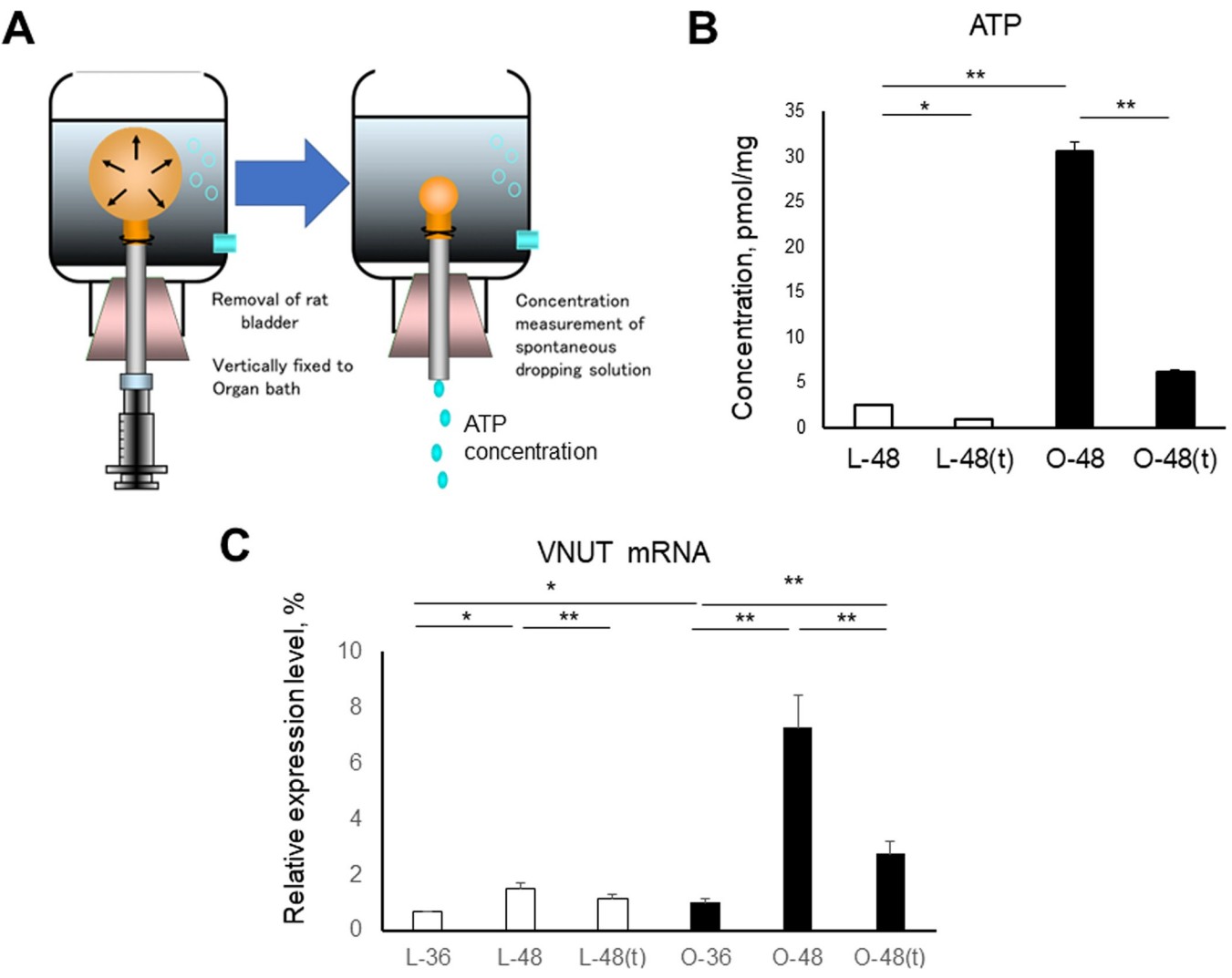

**Fig 2. Measurements of the amount of ATP released by bladder distention and VNUT expression in the bladder. A:** Organ bath bladder distention test. A whole bladder was set in an organ bath and injected with saline, which was collected using gravity after the syringe was removed. Republished from ref. No. 17 under a CC BY license, with permission from Osamu Yokoyama, original copyright in 2011. **B:** Changes in ATP levels released from the stretched bladder urothelium of LETO and OLETF rats at 48 weeks under distention, with 1.5 mL of Krebs solution at approximately 0.04 mL/s. **C:** Bladder VNUT mRNA levels relative to those at 36 weeks in LETO rats. Data are shown as the mean ± SEM. *p<0.05, **p<0.01.

## mRNA and protein expression of pro-inflammatory cytokines and growth factors in the bladders of OLETF rats

In OLETF rats (n = 3, each), we evaluated changes in pro-inflammatory cytokines (IL-6 and TNF-α) and growth factors (IGF-1 and bFGF) that are associated with bladder inflammation. The mRNA expression levels of IL-6, TNF-α, IGF-1, and bFGF were significantly higher at 48 weeks than at 36 weeks (p<0.01) and tended to be decreased by tadalafil treatment, although these apparent decreases were only significant for TNF-α (p<0.01) (Fig 5A). The protein expression levels of IL-6, TNF-α, IGF-1, and bFGF were also significantly higher at 48 weeks than at 36 weeks (p<0.01), and tended to be decreased by tadalafil treatment; these differences were significant for IL-6, TNF-α, and IGF-1 (p<0.05 or p<0.01). Together, these results indicate the altered expression levels of pro-inflammatory cytokines and growth factors in OLETF rats at an advanced age and with tadalafil treatment.

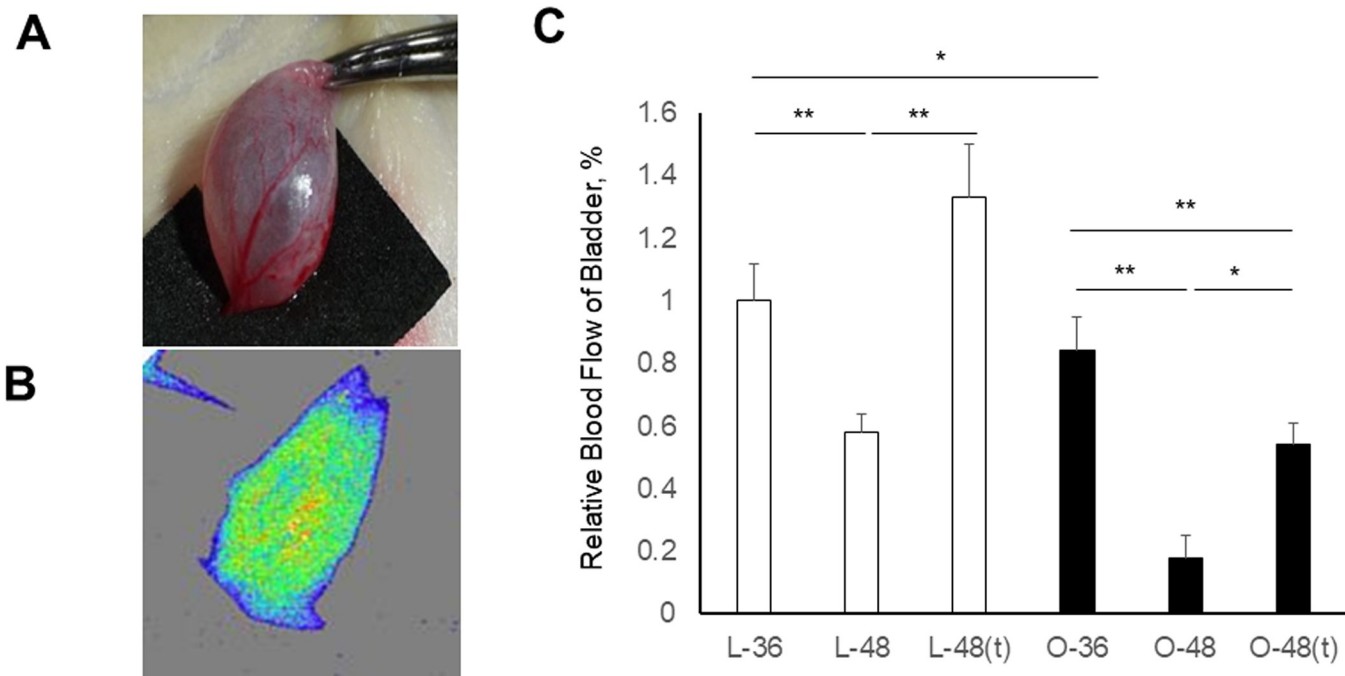

**Fig 3. Determination of bladder blood flow in LETO and OLETF rats. A:** Image of a rat bladder dissected and exposed under anesthesia. **B:** Laser speckle image of a rat bladder. **C:** Comparisons of bladder blood flow among the six rat groups. Blood flow is expressed as a percentage; the value of the L-36 rats was considered 100% (i.e., the levels are expressed relative to the levels at 36 weeks in LETO rats). Six samples were included in each group. Data are shown as the mean ± SEM. *p<0.05, **p<0.01.

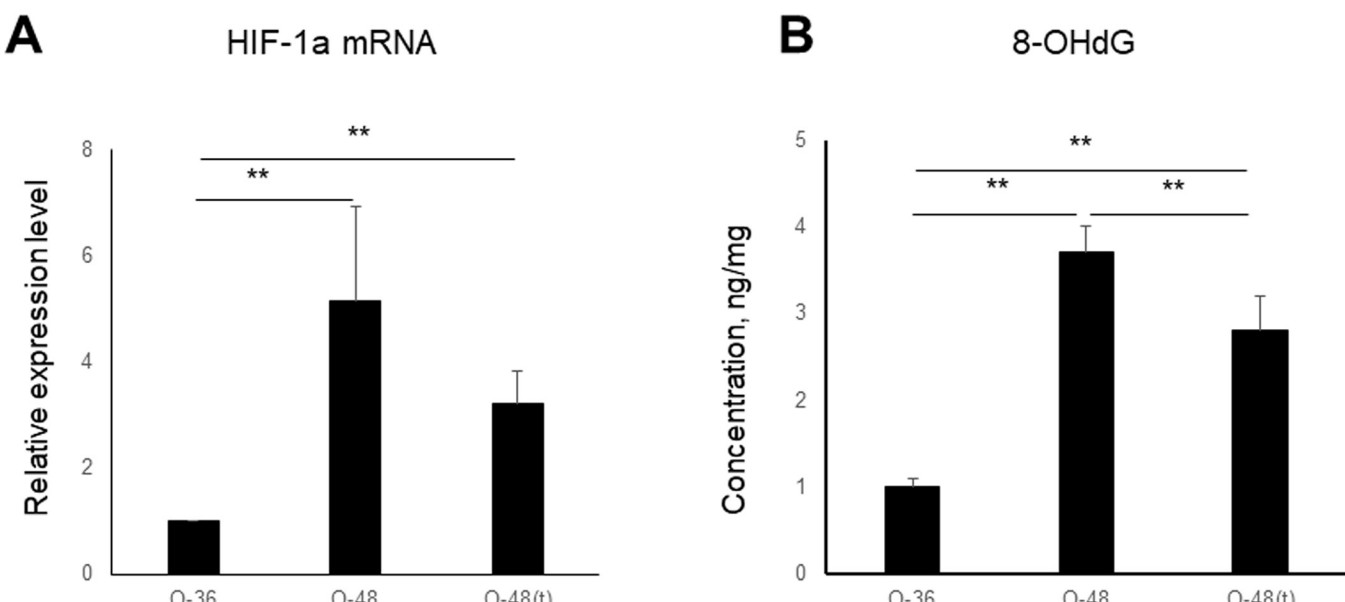

**Fig 4.** Changes in HIF-1α **(A)** and 8-OHdG **(B)** mRNA levels in the bladders of OLETF rats at 36 weeks and at 48 weeks with and without tadalafil. Six samples were included in each group, and two replicates were performed for each sample. Data are shown as the mean ± SEM. *p<0.05, **p<0.01.

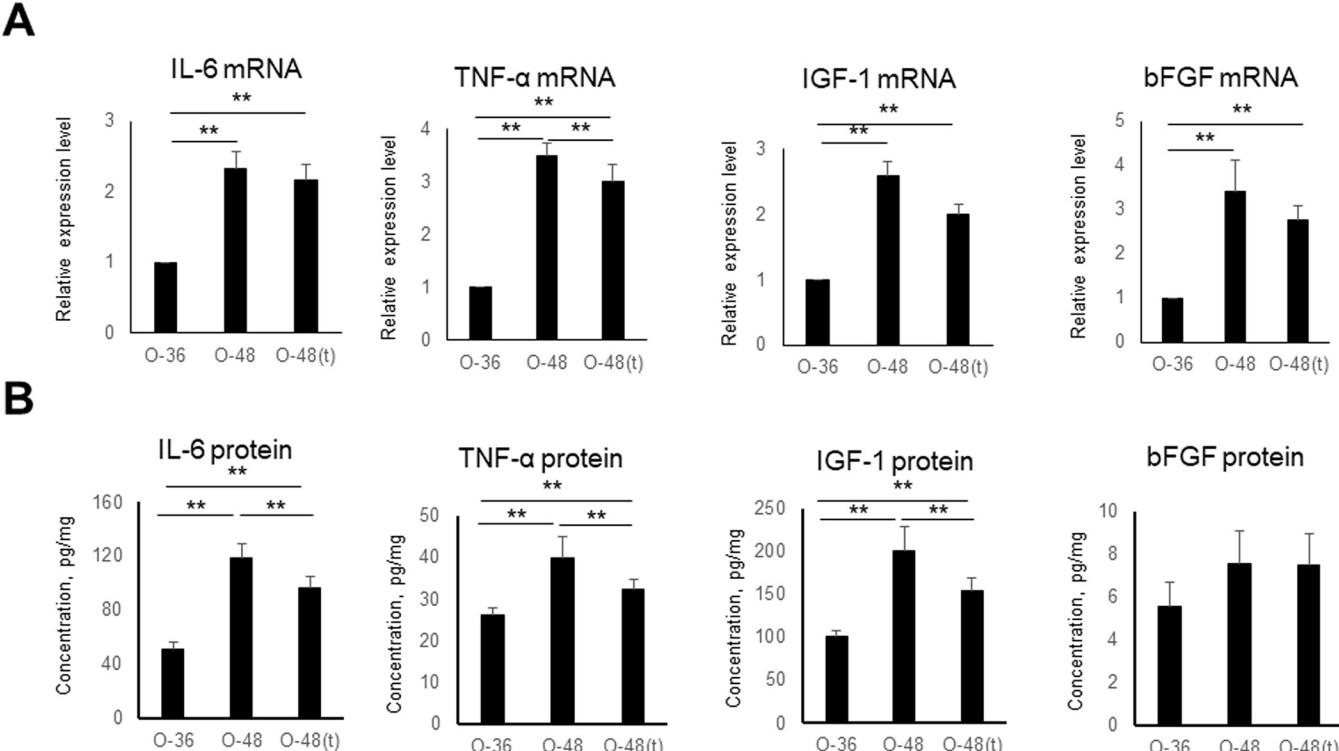

**Fig 5.** mRNA expression (relative to glyceraldehyde 3-phosphate dehydrogenase) **(A)** and protein concentrations **(B)** of pro-inflammatory cytokines (IL-6 and TNF-α) and growth factors (IGF-1 and bFGF) in the bladders of OLETF rats at 36 weeks and at 48 weeks with and without tadalafil, using real-time PCR and ELISA. Six samples were included in each group, and two replicates were performed for each sample. Data are shown as the mean ± SEM. *p<0.05, **p<0.01.

### Contractile responses of bladder strips to KCl, carbachol, and atropine in OLETF rats

We used bladder strips from OLETF rats (n = 3, each) to evaluate the bladder contraction responses to KCl and carbachol. Bladder contraction responses to KCl administration were significantly lower at 48 weeks than at 36 weeks (p<0.05). Furthermore, tadalafil treatment significantly increased bladder contractions (p<0.01) (Fig 6A). The dose–response curves of bladder contractions with a dose of carbachol revealed that contractions were increased by carbachol administration. Furthermore, sensitivity to carbachol was lower in the bladder at 48 weeks than at 36 weeks; this sensitivity in the bladder at 48 weeks was restored by tadalafil treatment (Fig 6B). Together, these results indicate that the bladder contractile responses of OLETF rats decrease with aging and are recovered by tadalafil treatment.

## Discussion

The results of the present study demonstrated that in a rat model of T2DM, the expression levels of HIF-1α, 8-OHdG, various pro-inflammatory cytokines and growth factors were increased as a result of bladder perfusion disorder. Moreover, our findings indicate that increased pro-inflammatory cytokines and growth factors may enhance the bladder urothelial release of ATP via VNUT upregulation. The bladder contractile response to KCl and muscarinic stimulation decreased with aging in the rats. Long-term (12-week) administration of tadalafil improved blood flow and oxidative stress, resulting in decreased ATP release from the urothelium and improved bladder storage and voiding function. The current study is the first

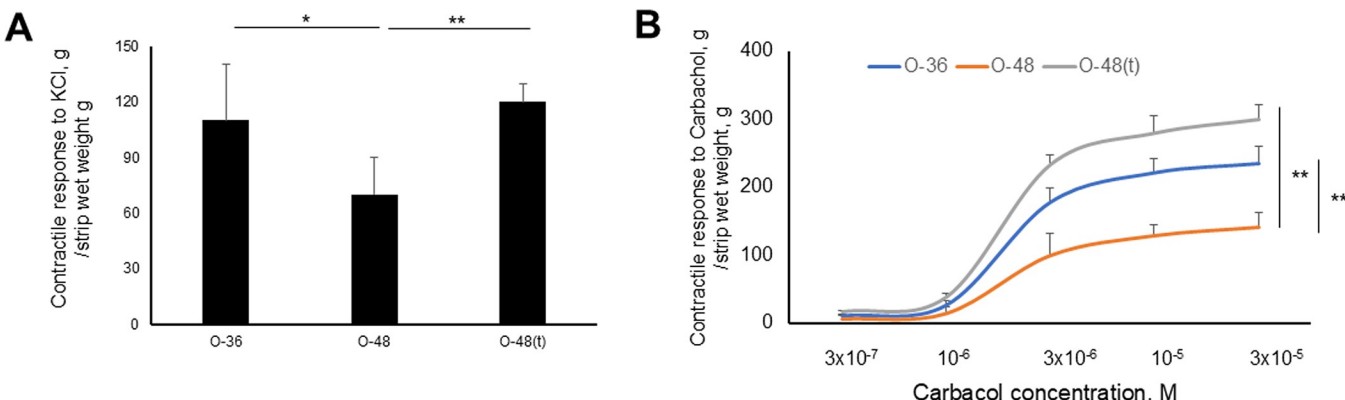

**Fig 6.** Contractile responses of bladder strips to 62 mM KCl **(A)**, carbachol ($3\times10^{-7}$, $10^{-6}$, $3\times10^{-6}$, $10^{-5}$, or $3\times10^{-5}$ M) **(B)** from OLETF rats at 36 weeks and at 48 weeks with and without tadalafil. Six samples were included in each group, and two replicates were performed for each sample. Data are shown as the mean ± SEM. *$p<0.05$, **$p<0.01$.

to suggest that urothelium-derived ATP may be associated with urine storage dysfunction in T2DM. It is also the first to indicate that tadalafil can inhibit ATP release and improve bladder contractions.

Diabetes can cause lower urinary tract dysfunction, including DO or DU [5, 19]. There are many reports of bladder function or morphology in animal models of streptozotocin-induced type 1 diabetes mellitus, most of which are characterized by the presence of bladder hypertrophy [20]. Two hereditary models of T2DM (namely, Zucker diabetic fatty rats and db/db mice) with blood glucose levels that are similar to those of the type 1 diabetes mellitus model show the same degree of, or only slight, hypertrophy [21]. By contrast, the Goto-Kakizaki rat—a hereditary model of T2DM—shows mild-to-moderate bladder hypertrophy [22]. Our T2DM model (i.e., OLETF rats) had an increased bladder wet weight with age, and bladder hypertrophy was observed compared with LETO rats. Moreover, the daily urine volume was significantly larger in OLETF rats than in LETO rats. Similarly, polyuria, caused by renal dysfunction, commonly appears in T2DM patients with aging. Diabetic polyuria induces the stimulation of DNA synthesis in the bladder, resulting in increased protein synthesis and hyperplasia of the smooth muscle and epithelial layers [23].

Although overactive bladder is observed from an early stage of T2DM in humans, the presence of DO has not yet been reported in a cystometric evaluation of a T2DM animal model [24]. In rats fed a fructose-rich diet, no change in bladder capacity was observed but increased non-voiding contractions during the storage phase were identified by cystometric recordings [25]; however, quantitative data of non-voiding contractions were not provided. In our OLETF model, the mean voided volume did not differ between LETO and OLETF rats, although non-voiding contractions were observed before the micturition reflex on cystometric evaluation. Total urine volume was associated with frequent micturition in OLETF rats. Moreover, our findings indicate that tadalafil might be able to suppress urine overproduction caused by T2DM. It has been reported that tadalafil treatment reduces glucose levels and has anti-inflammatory cardioprotective effects in leptin receptor-null diabetic mice [26]. Additionally, several preclinical and clinical studies have demonstrated the beneficial metabolic effects of PDE5 inhibitors for manifestations of metabolic syndrome [27, 28]. A randomized control study also showed the effects of high-dose tadalafil on decreasing hemoglobin A1c levels in patients with well-controlled T2DM [29]. However, no previous reports have shown the effects of PDE5 inhibitors on renal dysfunction caused by T2DM. Further experiments are therefore required to elucidate the association between PDE5 and renal function.

The present study is the first to demonstrate increased ATP release from the bladder urothelium in an animal T2DM model. Given that the PDE5 inhibitor improved bladder blood flow and decreased ATP release in the rats, it can be speculated that—at least in the present model—hypoxia triggers ATP release from the urothelium. Nephropathy, retinopathy, and neuropathy (the three major complications of T2DM) develop when vascular endothelial cells are exposed to hyperglycemia, subsequently causing microcirculatory disturbance. This condition develops into macrovascular complications, including coronary artery and cerebrovascular complications [30]. It has been reported that OLETF rats develop spontaneous persistent hyperglycemia with increased atherogenesis in arteries throughout the body before the onset of T2DM [31]. These atherosclerotic changes in the internal iliac and bladder feeding artery lead to bladder ischemia, resulting in increased levels of HIF-1α, 8-OHdG, and various pro-inflammatory cytokines and growth factors in the bladder wall. Several mechanisms have been postulated to underlie the release of ATP from the bladder urothelium during bladder distension, including vesicular exocytosis and connexin/pannexin channels [32–35]. It has also been suggested that ATP release is regulated mainly by VNUT (gene name: *Slc17A9*) in the bladder urothelium [15]. ATP accumulates and is stored in vesicles inside epithelial cells. The molecular machine VNUT is an active transporter that concentrates ATP into secretory vesicles [36]. The mild stretching of bladder urothelium harvested from VNUT-knockout mice reduces ATP release, suggesting that VNUT-mediated ATP release is involved in the urine storage mechanism that promotes bladder relaxation during the early stages of filling [37]. ATP release from the bladder urothelium is elevated in various human pathological conditions, such as overactive bladder, benign prostate hyperplasia, spinal cord injury, and interstitial cystitis [38–40]. We have also demonstrated that urothelium-derived ATP is increased in salt-sensitive hypertensive rats, with a corresponding decrease in the mean voided volume [7]. In the present study, bladder distention-evoked ATP release was significantly higher in OLETF rats than in LETO rats. In addition, VNUT expression was markedly elevated in the bladders of OLETF rats. There have been few reports of the enhancement of ATP release by inflammation. Mice with conditional hepatic double-knockout of *Irs1* and *Irs2* show characteristics of T2DM and overactive bladder [41]. In this animal model, when TNF-α-mediated signaling (which was increased in the serum and bladder) was inhibited, overactive bladder was reversed without affecting hyperglycemia. In the present study, the expression levels of IL-6, TNF-α, IGF-1, and bFGF in the bladders of OLETF rats increased with age and decreased with tadalafil administration. Further investigations are therefore necessary to elucidate the association between inflammation and ATP release in the bladder urothelium.

Although the nitric oxide/cyclic guanosine monophosphate (cGMP) pathway is an endothelium-derived signaling pathway that induces vascular smooth muscle relaxation, it is suggested that this pathway is also involved in bladder relaxation. PDE5 inhibitor, sildenafil, increases cGMP levels and inhibits the distension-induced release of ATP from the bladder urothelium in mice [42]. Another study indicated that sildenafil inhibits ATP release from mucosa isolated from the detrusor muscle [43]. Together, these findings suggest that the nitric oxide/cGMP pathway may suppress epithelial ATP release. A possible mechanism by which PDE5 inhibitors might suppress ATP release involves the attenuation of $Ca^{2+}$ influx via transient receptor potential vanilloid 2 and 4 [44]. In the present study, 12 weeks of tadalafil administration markedly decreased ATP release from the bladder urothelium. It is unclear whether this was an acute effect via the nitric oxide/cGMP pathway or a chronic effect associated with decreased pro-inflammatory cytokines as a result of improved blood flow. However, given that the effect occurred >2 days after the completion of the 12-week tadalafil treatment, and because the insides of the bladders were rinsed three times with 0.5 mL Krebs solution, we

speculate that the effect may have been the result of improved bladder blood flow caused by the 12-week tadalafil regimen.

We also observed that the contractile responses to KCl and carbachol were significantly lower in bladder strips from 48-week-old OLETF rats than in those from 36-week-old OLETF rats, suggesting that muscarinic receptor hyposensitivity and/or decreased contractile factors associated with increased collagen deposition in bladder smooth muscle occurs during the late stage of T2DM. Although we did not conduct a histological examination, the bladder is known to become less responsive to muscarinic stimulation, which is accompanied by bladder hypertrophy. One of the later stages of diabetic bladder dysfunction has been described as voiding dysfunction arising from an underactive bladder [5]. This is presumed to be caused by the accumulation of oxidative stress products as the result of long-lasting hyperglycemia [45]. In rats, chronic bladder ischemia produced by endothelial injury of the iliac arteries decreases the contractile responses of bladder strips to KCl, electrical field stimulation, and carbachol, with collagen deposition in the smooth muscle layer. In this previous study, when tadalafil was administered for 8 weeks concurrently with the endothelial injury, both the decline in bladder contractility and collagen deposition were prevented. Similarly, our present findings in rats indicate that tadalafil administration starting at 36 weeks can prevent T2DM-associated bladder contraction disorders. PDE5 inhibitors may thus be useful therapeutic tools for voiding dysfunction associated with T2DM.

## Limitations

The difference in the voiding frequency in LETO and OLETF rats might be mainly caused by the difference in total urine volume. Based on the results that tadalafil treatment decreased the urine volume in OLETF rats, it is suggested that tadalafil is effective for polyuria caused by T2DM. The changes in the total urine volume might be associated with the renal deficiency caused by T2DM. However, the mechanisms for them were not evaluated in this study. We are planning to make experiments in the future study.

To explore bladder function, we conducted in vitro contraction experiments of the detrusor muscle but did not perform histological examinations of the bladder wall. It is therefore not possible to say with certainty why the contractions in response to carbachol were reduced. Furthermore, an important question remains: why does T2DM cause increased ATP release from the bladder urothelium? Although pro-inflammatory cytokines have been suggested as the responsible candidates, cytokine-specific inhibitors need to be used to test this possibility.

## Conclusions

Using OLETF rats, a series of pathological conditions associated with T2DM were reproduced, including polyuria, bladder wall ischemia, increases in pro-inflammatory cytokines and growth factors, epithelial-mediated ATP release, and impaired bladder contractility. Tadalafil, a PDE5 inhibitor, improved polyuria, bladder ischemia, reduced the levels of pro-inflammatory cytokines and growth factors, inhibited ATP release from the bladder urothelium, and improved bladder contractility. This pathological model will continue to help to elucidate the pathology of renal and bladder dysfunction associated with T2DM and contribute to the design of new therapeutic strategies.

## Supporting information

**S1 Table. Primers sequences of the genes examined by real-time PCR.**
(TIF)

## Acknowledgments

We thank all of the project members. We also thank Mark Cleasby, PhD, and Bronwen Gardner, PhD, from Edanz (https://jp.edanz.com/ac) for editing a draft of this manuscript.

## Author Contributions

**Conceptualization:** Takafumi Kabuto, Minekatsu Taga, Masaya Seki, Osamu Yokoyama, Naoki Terada.

**Data curation:** Takafumi Kabuto, Hisato Kobayashi, Xinmin Zha, Keiko Nagase, Nobuki Tanaka, Yoshinaga Okumura.

**Formal analysis:** Takafumi Kabuto, So Inamura, Osamu Yokoyama.

**Funding acquisition:** Osamu Yokoyama, Naoki Terada.

**Investigation:** Takafumi Kabuto, Xinmin Zha, Keiko Nagase, Nobuki Tanaka, Yoshinaga Okumura.

**Methodology:** Takafumi Kabuto, Xinmin Zha, Keiko Nagase, Nobuki Tanaka, Yoshinaga Okumura.

**Project administration:** So Inamura, Minekatsu Taga, Masaya Seki, Osamu Yokoyama, Naoki Terada.

**Resources:** Takafumi Kabuto, Osamu Yokoyama.

**Software:** So Inamura, Minekatsu Taga, Masaya Seki, Naoki Terada.

**Supervision:** So Inamura, Minekatsu Taga, Masaya Seki, Osamu Yokoyama, Naoki Terada.

**Validation:** Hisato Kobayashi, Minekatsu Taga, Masaya Seki, Osamu Yokoyama, Naoki Terada.

**Visualization:** Hisato Kobayashi, Osamu Yokoyama.

**Writing – original draft:** Takafumi Kabuto.

**Writing – review & editing:** So Inamura, Hisato Kobayashi, Xinmin Zha, Keiko Nagase, Minekatsu Taga, Masaya Seki, Nobuki Tanaka, Yoshinaga Okumura, Osamu Yokoyama, Naoki Terada.

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
