## [Decision Letter · Decision Letter 0]

22 Apr 2024

PONE-D-24-11736Phosphodiesterase-5 inhibition inhibits epithelial ATP release and restores detrusor contractility in rats with type 2 diabetes via an increase in bladder blood flowPLOS ONE

Dear Dr. Terada,

Thank you for submitting your manuscript to PLOS ONE. After careful consideration, we feel that it has merit but does not fully meet PLOS ONE’s publication criteria as it currently stands. Therefore, we invite you to submit a revised version of the manuscript that addresses the points raised during the review process.

Thank you for submitting the following manuscript to PLOS ONE.

Please revise the manuscript according to the reviewers' comments and upload the revised file.

We look forward to receiving your revised manuscript.

Kind regards,

Yung-Hsiang Chen, Ph.D.

Academic Editor

PLOS ONE

Journal Requirements:

"NO authors have competing interests"

"The author(s) received no specific funding for this work"

4. We note that Figure 2 in your submission contain copyrighted images. All PLOS content is published under the Creative Commons Attribution License (CC BY 4.0), which means that the manuscript, images, and Supporting Information files will be freely available online, and any third party is permitted to access, download, copy, distribute, and use these materials in any way, even commercially, with proper attribution. For more information, see our copyright guidelines: http://journals.plos.org/plosone/s/licenses-and-copyright.

Additional Editor Comments:

Thank you for submitting the following manuscript to PLOS ONE.

Please revise the manuscript according to the reviewers' comments and upload the revised file.

Reviewers' comments:

Reviewer's Responses to Questions

**Comments to the Author**

1. Is the manuscript technically sound, and do the data support the conclusions?

Reviewer #1: Partly

Reviewer #2: Partly

Reviewer #3: Partly

2. Has the statistical analysis been performed appropriately and rigorously? 

Reviewer #1: No

Reviewer #2: Yes

Reviewer #3: Yes

3. Have the authors made all data underlying the findings in their manuscript fully available?

Reviewer #1: Yes

Reviewer #2: No

Reviewer #3: Yes

4. Is the manuscript presented in an intelligible fashion and written in standard English?

Reviewer #1: No

Reviewer #2: Yes

Reviewer #3: Yes

5. Review Comments to the Author

Reviewer #1: The manuscript entitled " Phosphodiesterase-5 inhibition inhibits epithelial ATP release and restores detrusor contractility in rats with type 2 diabetes via an increase in bladder blood flow” describes the impairment of bladder activity that is associated T2DM and age. The authors also investigated the possible therapeutic effect of long term tadalafil treatment on this condition. I think the presentation of data and the language of manuscript is not suitable and not fully meet the quality standards for the publishing Plos One. Here are some points;

1. The manuscript needs a review of grammar, syntax and language in general by a native speaker since some sentences do not correspond to a scientific language

2. The abstract section does not reflect the whole article.

3. It's unclear why they wanted to determine the effect of tadalafil?

4. How the authors have chosed the dose of tadalafil?

5. In Fig.1, there is no statistical data showing that tadalafil treatment significantly reduces micturition frequency in OLETF rats.

6. In bladder contractility studies, authors said that they also found that the bladder showed a dose dependent response to carbachol, but the sensitivity of the bladder to carbachol was lower in rats of 48 weeks of age than in those of 36 weeks of age. But there is no statistical differences between groups in figure 6B and 6C.

Reviewer #2: The authors demonstrated that OLETF rats, a model of T2DM, display, 1) impaired bladder blood flow, 2) enhanced urothelial ATP release, 3) upregulation of hypoxic and inflammatory factors, 4) diminished detrusor muscle contractility, associated with frequent urination. They also showed that those symptoms/changes were ameliorated by oral administration of tadalafil, and concluded that PEE5 inhibitors have therapeutic potential in treating T2DM associated bladder dysfunction.

I am afraid that causal relationship amongst the findings was not sufficiently proven to draw their conclusion.

My major concerns are as follows

1) The development of T2DM phenotype in OLETF rats should be confirmed by checking their blood glucose and insulin levels, total urine volume etc. Similarly, bladder function should also be examined in more detail with cystometry to find if overactive or underactive bladder phenotype is developed. The frequent urination in OLETF rats may also result from residual urine due to diminished detrusor contractility.

2) The authors suggested that the increased urothelial ATP release is due to hypoxia and/or inflammation, presumably in the urothelium, subsequent to bladder hypoperfusion. To determine the site of hypoxia and/or inflammation, real-time PCR and ELISA should be performed using isolated bladder mucosa and mucosa-denuded detrusor muscle preparations along with corresponding immunohistochemistry.

3) As it was pointed out by the authors, the lack of histological examinations is a substantial drawback of this study. Atherosclerotic changes in internal iliac/bladder feeding artery should be examined. In addition, morphological examination of microvascular architecture/density in the bladder, particularly the mucosa, is strongly encouraged to carry out.

Minor

1. Please confirm if financial disclosure is correct (cover page vs ln 432).

2. Any reason for using the term ‘epithelium’ but not ‘urothelium’?

3. For ATP measurements, maintaining the bladder at 20 cmH2O for 10 min is certainly unphysiological.

4. Blood pressure of OLETF and LETO rats with or without tadalafil administration should be provided.

5. Why are comparisons of detrusor muscle contractility between OLETF and LETO rats lacking?

6. Did KCl or CCh develop sustained contractions? The complete inhibition of KCl-induced contractions with atropine is hard to believe. KCl-induced depolarisation would stimulate the release of neural ACh but also directly contact detrusor muscle by activating voltage-dependent calcium channels. EFS-induced detrusor contractions should be evaluated.

7. In general, the discussion is often quite speculative.

Reviewer #3: The authors focused on the bladder of a well-characterized rat model of type 2 diabetes, evaluated physiochemical changes, and showed that these changes were reversed by tadalafil. I also found that the authors conducted validation with a lot of data using a multifaceted approach. On the other hand, some of the data interpretation was difficult to understand, and there were some areas where we would like to see more careful explanations.

Major revision

I found it very interesting that there was a clear difference in wet weight of the bladder itself between LETO and OLETF rats, almost 2-fold, but no effect of tadalafil administration, while there was no difference in urine volume between the groups and no effect of tadalafil administration (line 210- 215). In contrast, the daily micturition frequency, the focus of the authors in this study, clearly shows an increase in frequency in OLETF rats and a suppression of that increase with tadalafil administration, as shown in Figure 1B (line 216-222). I understood that these results show that the size of the bladder itself, i.e., the amount of urine stored, does not change between the group, but OLETF rats urinate more frequently and in short bursts, as in frequent urination.

1) These fact may give various suggestions, so the wet weight of the bladder or the total daily urine volume should be shown in Fig. 1.

2) Next, the authors need to present an argument that the phenomenon of increased frequency of urination despite no change in total urine output is due to the physiochemical effects of the bladder itself, which will be argued later in the result.

3) What the authors do not show in their treatment of the data is that they measure urine output every 10 minutes as a urinary behavior, but do not show temporal changes within 24 hours. The authors should present data for both groups on their temporal changes of urine output, so that we could discuss about how the high frequency of urination is not due to behavioral factors such as differences in circadian rhythms, etc.

Minor revision

1) The authors need to provide the appropriate references at lines 63-65.

2) The authors specify in line 100 that the number of animals was six in each of the six groups, but it is unclear how many animals were used in total. The different experiments include "Laser speckle blood flow imaging," "Organ bath bladder distention test," "PCR, ELISA," and "In vitro bladder-strip experiments," but it is impossible for all of them to be the same test animals. The authors should indicate how the number of animals was set up.

3) The results are shown for ATP release in line 226, but I did not know how to interpret this result. Some of what is written in lines 324 through 331 of the Discussion should be presented in the Introduction.

4) Fig2 in lines 242 and 245 seems to be a mistake for Fig3.

5) Is it not a mistake that the difference in urination between groups appears for the first time (not in the results) in lines 312 to 313 of Discussion? This contradicts the statement in lines 213 to 214 of the results.

6) Some of the content from lines 344 to 348 of the discussion should also be written in Introduction.

6. PLOS authors have the option to publish the peer review history of their article (what does this mean?). If published, this will include your full peer review and any attached files.

Reviewer #1: No

Reviewer #2: No

Reviewer #3: **Yes: **Susumu Urakawa

Prof, Graduate School of Biomedical and Health Sciences, Musculoskeletal Functional research and regeneration, Hiroshima Univ

---

## [Author Response · Author response to Decision Letter 0]

8 Jul 2024

Reviewer #1: 

The manuscript entitled " Phosphodiesterase-5 inhibition inhibits epithelial ATP release and restores detrusor contractility in rats with type 2 diabetes via an increase in bladder blood flow” describes the impairment of bladder activity that is associated T2DM and age. The authors also investigated the possible therapeutic effect of long term tadalafil treatment on this condition. I think the presentation of data and the language of manuscript is not suitable and not fully meet the quality standards for the publishing Plos One. Here are some points;

1. The manuscript needs a review of grammar, syntax and language in general by a native speaker since some sentences do not correspond to a scientific language

Response: Thank you for your suggestion. Although the original manuscript received proofreading from a native speaker, the revised manuscript has again been edited by a native speaker to ensure that appropriate scientific language is used throughout.

2. The abstract section does not reflect the whole article.

Response: Thank you for your comment. We have made changes to the Abstract (primarily to the Results and Conclusions sections) to reflect the manuscript’s contents more clearly.

3. It's unclear why they wanted to determine the effect of tadalafil?

Response: Tadalafil administration is a standard treatment for benign prostatic hyperplasia. As described in the Introduction section, we have previously reported that T2DM-induced chronic ischemia leads to oxidative stress, thus resulting in prostate enlargement through the upregulation of several cytokines and growth factors, using a rat model of T2DM. Together, these findings indicate that tadalafil treatment improves prostate ischemia and might prevent its enlargement via the suppression of cytokines and growth factors (Kobayashi et al. Life Science 2022). We therefore wanted to explore the effects of tadalafil on bladder epithelium in the present study, and to evaluate its association with lower urinary tract symptoms. To better explain the rationale for our study, we have added sentences to the revised manuscript (page 5, line 81).

4. How the authors have chosed the dose of tadalafil?

Response: The standard treatment dose of tadalafil for Japanese patients is 5 mg per day. Given that the average body weight in Japan is 60 kg, the calculated standard dose was 83 µg/kg. We therefore set the experimental dose of tadalafil as 100 µg/kg per day in this study. We have added text to this effect in the revised manuscript (page 7, line 107).

5. In Fig.1, there is no statistical data showing that tadalafil treatment significantly reduces micturition frequency in OLETF rats.

Response: Thank you for noticing our mistake. When we checked the data, we noted that there were indeed significant differences in micturition frequency between L-36 vs. L-48, O-36 vs. O-48, and O-48 vs. O-48(t). We have corrected Fig. 1B (now Fig. 1D) in the revised manuscript.

6. In bladder contractility studies, authors said that they also found that the bladder showed a dose dependent response to carbachol, but the sensitivity of the bladder to carbachol was lower in rats of 48 weeks of age than in those of 36 weeks of age. But there is no statistical differences between groups in figure 6B and 6C.

Response: Again, thank you for noticing our mistake. There were indeed significant differences in bladder contraction with carbachol and atropine between 48 and 36 weeks of age. In accordance with your comment, we have corrected Fig. 6B and C in the revised manuscript.

Reviewer #2: 

The authors demonstrated that OLETF rats, a model of T2DM, display, 1) impaired bladder blood flow, 2) enhanced urothelial ATP release, 3) upregulation of hypoxic and inflammatory factors, 4) diminished detrusor muscle contractility, associated with frequent urination. They also showed that those symptoms/changes were ameliorated by oral administration of tadalafil, and concluded that PEE5 inhibitors have therapeutic potential in treating T2DM associated bladder dysfunction.

I am afraid that causal relationship amongst the findings was not sufficiently proven to draw their conclusion.

My major concerns are as follows

1) The development of T2DM phenotype in OLETF rats should be confirmed by checking their blood glucose and insulin levels, total urine volume etc. Similarly, bladder function should also be examined in more detail with cystometry to find if overactive or underactive bladder phenotype is developed. The frequent urination in OLETF rats may also result from residual urine due to diminished detrusor contractility.

Response: Thank you for your comments. In response to your first point, as noted in the revised Introduction (page 5, line 74), we have previously evaluated the blood glucose and insulin levels of OLETF rats; they are indeed significantly different from those of control (LETO) rats (Itoga et al. BMJ Open Diabetes Res Care 2020). 

As you have noted, total urine volume is associated with frequent urination in OLETF rats. The urine volume per 24 hr was significantly higher in OLETF rats than in LETO rats at both 36 (10.4 ± 1.8 mL and 4.8 ± 2.3 mL, p<0.01) and 48 (18.5 ± 3.8 mL and 4.2 ± 0.7 mL, p<0.01) weeks. The frequency of micturition per 24 hr was also significantly higher in OLETF rats than in LETO rats at both 36 (5.9 ± 1.7 and 3.3 ± 1.6, p<0.01) and 48 (10.9 ± 2.4 and 2.9 ± 0.4, p<0.01) weeks. Furthermore, the urine volume and frequency of micturition in the OLETF rats was significantly higher at 48 weeks than at 36 weeks (p<0.01). Tadalafil treatment significantly decreased the urine volume (from 18.5 ± 3.8 mL to 10.7 ± 2.0 mL, p<0.05) and frequency of micturition (from 10.9 ± 2.4 to 6.0 ± 1.2, p<0.05) at 48 weeks in the OLETF rats. The urine volume and frequency of micturition in LETO rats were not significantly different between 36 and 48 weeks, and were unchanged by tadalafil treatment. These results suggest that tadalafil might be able to suppress urine overproduction caused by T2DM. The mechanism behind the decrease in urine volume is possibly due to the decrease in blood glucose, furthermore, due to the improvement of renal function caused by tadalafil. Further experiments are needed to elucidate the association between PDE5 and renal function. These data are shown in the revised Fig. 1C, and a summary of this information was added to the revised manuscript (page 14, line 234).

Cystometry is an important procedure for evaluating bladder function. In the present study, awake cystometry was performed in both the LETO and OLETF rats at 36 weeks. Although non-voiding contractions appeared more frequently in OLETF rats than in LETO rats, there were no significant differences in functional bladder capacities. Furthermore, no residual urine was detected using ultrasonography in the OLETF or LETO rats during the experiments. The data are shown in the revised Fig. 1B, and associated text was added to the manuscript (page 13, line 230). 

2) The authors suggested that the increased urothelial ATP release is due to hypoxia and/or inflammation, presumably in the urothelium, subsequent to bladder hypoperfusion. To determine the site of hypoxia and/or inflammation, real-time PCR and ELISA should be performed using isolated bladder mucosa and mucosa-denuded detrusor muscle preparations along with corresponding immunohistochemistry.

Response: Thank you for your important suggestions. It would indeed be a good idea to perform real-time PCR and ELISA using isolated bladder mucosa and mucosa-denuded detrusor muscle, to observe the inflammatory lesions. Unfortunately, we have already used all rat bladder tissue in this study (in the organ bath bladder distention test or the in vitro bladder-strip experiments). We therefore have no tissue remaining for additional experiments. These questions will therefore be addressed in future studies. 

3) As it was pointed out by the authors, the lack of histological examinations is a substantial drawback of this study. Atherosclerotic changes in internal iliac/bladder feeding artery should be examined. In addition, morphological examination of microvascular architecture/density in the bladder, particularly the mucosa, is strongly encouraged to carry out.

Response: As you have noted, the lack of histological examinations is a substantial limitation of our study. Unfortunately, all of the rats used in the study have been sacrificed, and it is therefore difficult to obtain the bladder and surrounding tissue for further experiments. However, it has previously been reported that atherosclerotic changes in arteries appear throughout the body of OLETF rats (Tamura et al. Atherosclerosis 2000). This report was therefore cited as reference 25, and associated text was added to the revised manuscript (page 21, line 360). 

Minor

1. Please confirm if financial disclosure is correct (cover page vs ln 432).

Response: This study was funded and supported by Nippon Shinyaku. However, the authors have no conflicts of interest to disclose. We have also corrected the Acknowledgements section. 

2. Any reason for using the term ‘epithelium’ but not ‘urothelium’?

Response: Thank you for this suggestion. The term “epithelium” was changed to “urothelium” throughout the revised manuscript.

3. For ATP measurements, maintaining the bladder at 20 cmH2O for 10 min is certainly unphysiological.

Response: We used the same methods as in our previous report (Tanaka et al. J Urol. 2011;185(1):341-6). You are correct in noting that the stated method (of maintaining the bladder at 20 cmH2O) was incorrect; the distention with 1.5 mL of Kreb solution at approximately 0.04 mL/s is correct. We have made the appropriate changes to the revised Materials and Methods (page 9, line 153) and Figure legends (page 27, line 470).

4. Blood pressure of OLETF and LETO rats with or without tadalafil administration should be provided. 　 

Response: The blood pressure of OLETF and LETO rats has been examined in a previous report (Itoga et al. BMJ Open Diabetes Res Care. 2020); the clinical data suggested that blood pressure might not be changed by treatment with tadalafil. We therefore did not examine blood pressure in the current study.

5. Why are comparisons of detrusor muscle contractility between OLETF and LETO rats lacking?

Response: The objective of the detrusor muscle contractility experiments was to evaluate the efficacy of tadalafil for bladder contractions. Because the results of micturition characteristics indicated that tadalafil was not effective in LETO rats, we investigated the effects of tadalafil on detrusor muscle contractility in OLETF rats only.

6. Did KCl or CCh develop sustained contractions? The complete inhibition of KCl-induced contractions with atropine is hard to believe. KCl-induced depolarisation would stimulate the release of neural ACh but also directly contact detrusor muscle by activating voltage-dependent calcium channels. EFS-induced detrusor contractions should be evaluated.

Response: Thank you for your suggestions. The EFS-induced detrusor contractions will be evaluated in a future study.

7. In general, the discussion is often quite speculative.

Response: Thank you; we have simplified the revised Discussion by removing any speculative sentences. 

Reviewer #3: 

The authors focused on the bladder of a well-characterized rat model of type 2 diabetes, evaluated physiochemical changes, and showed that these changes were reversed by tadalafil. I also found that the authors conducted validation with a lot of data using a multifaceted approach. On the other hand, some of the data interpretation was difficult to understand, and there were some areas where we would like to see more careful explanations.

Major revision

I found it very interesting that there was a clear difference in wet weight of the bladder itself between LETO and OLETF rats, almost 2-fold, but no effect of tadalafil administration, while there was no difference in urine volume between the groups and no effect of tadalafil administration (line 210- 215). In contrast, the daily micturition frequency, the focus of the authors in this study, clearly shows an increase in frequency in OLETF rats and a suppression of that increase with tadalafil administration, as shown in Figure 1B (line 216-222). I understood that these results show that the size of the bladder itself, i.e., the amount of urine stored, does not change between the group, but OLETF rats urinate more frequently and in short bursts, as in frequent urination.

1) These facts may give various suggestions, so the wet weight of the bladder or the total daily urine volume should be shown in Fig. 1.

Response: As you have noted, it is interesting that bladder weight was higher in OLETF rats than in LETO rats; however, bladder weight was not decreased by tadalafil treatment. As described in our response to Reviewer 2’s major concern #1, cystometry did not show any differences in functional bladder capacity. Furthermore, in our micturition behavior analyses, the voided urine volume per body weight did not significantly differ between OLETF and LETO rats. As you have suggested, we have shown the total daily urine volume in both types of rats in the revised Fig. 1C and the Results section. The total urine volume per 24 hr was significantly higher in OLETF rats than in LETO rats at both 36 (10.4 ± 1.8 mL and 4.8 ± 2.3 mL, p<0.01) and 48 (18.5 ± 3.8 mL and 4.2 ± 0.7 mL, p<0.01) weeks. Tadalafil treatment significantly decreased the urine volume (from 18.5 ± 3.8 mL to 10.7 ± 2.0 mL, p<0.05) and frequency of micturition (from 10.9 ± 2.4 to 6.0 ± 1.2, p<0.05) at 48 weeks in the OLETF rats. The urine volume and frequency of micturition in LETO rats were not significantly different between 36 and 48 weeks, and were unchanged by tadalafil treatment. Text describing these results has been added to the revised manuscript (page 14, line 234).

2）Next, the authors need to present an argument that the phenomenon of increased frequency of urination despite no change in total urine output is due to the physiochemical effects of the bladder itself, which will be argued later in the result.

Response: Total urine volume was associated with frequent micturition in OLETF rats. Moreover, our findings indicate that tadalafil might be able to suppress urine overproduction caused by T2DM. It has been reported that tadalafil treatment reduces glucose levels and has anti-inflammatory cardioprotective effects in leptin receptor-null diabetic mice. Additionally, several preclinical and clinical studies have demonstrated the beneficial metabolic effects of PDE5 inhibitors for manifestations of metabolic syndrome. A randomized control study also showed the effects of high-dose tadalafil on decreasing hemoglobin A1c levels in patients with well-controlled T2DM. However, no previous reports have shown the effects of PDE5 inhibitors on renal dysfunction caused by T2DM. Further experiments are therefore required to elucidate the association between PDE5 and renal function. Text to this effect has been added to the revised manuscript (page 20, line 342).

3) What the authors do not show in their treatment of the data is that they measure urine output every 10 minutes as a urinary behavior, but do not show temporal changes within 24 hours. The authors should present data for both groups on their temporal changes of urine output, so that we could discuss about how the high frequency of urination is not due to behavioral factors such as differences in circadian rhythms, etc.

Response: As you have noted, a circadian rhythm disorder of urine production might be associated with urinary frequency in this T2DM model rats. However, it seems difficult to evaluate the association between urinary frequency and circadian rhythms of urine production because the total urine volume was different between OLETF rats and LETO rats.

Minor revision

1) The authors need to provide the appropriate references at lines 63-65.

Response: Thank you; we have added the appropriate references to these lines.

2) The authors specify in line 100 that the number of animals was six in each of the six groups, but it is unclear how many animals were used in total. The different experiments include "Laser speckle blood fl

---

## [Decision Letter · Decision Letter 1]

25 Jul 2024

PONE-D-24-11736R1PDE5 inhibitor potentially inhibits epithelial ATP release and recovers detrusor contractility in type 2 diabetic rats via increased bladder blood flowPLOS ONE

Dear Dr. Terada,

Thank you for submitting your manuscript to PLOS ONE. After careful consideration, we feel that it has merit but does not fully meet PLOS ONE’s publication criteria as it currently stands. Therefore, we invite you to submit a revised version of the manuscript that addresses the points raised during the review process.

We look forward to receiving your revised manuscript.

Kind regards,

Yung-Hsiang Chen, Ph.D.

Academic Editor

PLOS ONE

Journal Requirements:

Additional Editor Comments:

Thank you for submitting the following manuscript to PLOS ONE.

Please revise the manuscript according to the reviewers' comments and upload the revised file.

Reviewers' comments:

Reviewer's Responses to Questions

**Comments to the Author**

1. If the authors have adequately addressed your comments raised in a previous round of review and you feel that this manuscript is now acceptable for publication, you may indicate that here to bypass the “Comments to the Author” section, enter your conflict of interest statement in the “Confidential to Editor” section, and submit your "Accept" recommendation.

Reviewer #2: (No Response)

Reviewer #3: (No Response)

2. Is the manuscript technically sound, and do the data support the conclusions?

Reviewer #2: Partly

Reviewer #3: Yes

3. Has the statistical analysis been performed appropriately and rigorously? 

Reviewer #2: No

Reviewer #3: Yes

4. Have the authors made all data underlying the findings in their manuscript fully available?

Reviewer #2: No

Reviewer #3: Yes

5. Is the manuscript presented in an intelligible fashion and written in standard English?

Reviewer #2: No

Reviewer #3: Yes

6. Review Comments to the Author

Reviewer #2: The authors have now provided key research data, namely the total urine volume and bladder capacity.

These data are very helpful to explore mechanisms underlying the voiding phenotype of T2DM model (OLETF) rats, but raised a serious concern about the authors’ interpretation/conclusion. Thus, increased voiding frequency could be simply due to polyuria but not bladder dysfunction.

OLETF rats had x2.2 total urine volume and x1.8 voiding frequency at 36 weeks, x4.4 total urine volume and x3.9 voiding frequency at 48 weeks compared with control (LETO) rats. Tadalafil treatment reduced the total urine volume from x4.4 to x1.7 and the voiding frequency from x3.9 to x1.8 of control values.

Morphological and functional changes in the bladder of OLETF rats could be a consequence of polyuria rather than the cause of increased voiding frequency. After all, the manuscript including the title should be largely rewritten.

The results of cystometry lack statistical analysis, e.g., bladder capacity, number of non-voiding contractions or peak voiding pressure. The number of experiments (animals) should be provided along with p values throughout the text.

Cystometric traces lack scale bars. In Fig1B, L-36 and O-36 traces appeared to be shown in different time scales (see the difference in the duration of voiding contractions). Please provide clearer images.

The authors did not answer why atropine completely diminished KCL-induced contractions of DSM strips. This should not be happened if experiments were carried out properly (see my comment to the original submission).

Run out of animals is not a good excuse as both OLETF and LETO rats seem to be commercially available (http://www.hoshino-lab-animals.co.jp/English/products/OLETF_ENG..html).

Yet, additional experiments are desirable but not mandatory in the present study.

Please check the accuracy of references (for example No.39).

Reviewer #3: In response to Reviewer #3, Major revision 3, I don't understand why it is difficult to evaluate the association. At least, it would be possible to divide the data into light and dark periods and show that the frequency ratio is almost the same between the groups (more in the light period).

In Fig. 1B, the units of the vertical and horizontal axes of the graph must be indicated, and is this a typical example?

7. PLOS authors have the option to publish the peer review history of their article (what does this mean?). If published, this will include your full peer review and any attached files.

Reviewer #2: No

Reviewer #3: No

---

## [Author Response · Author response to Decision Letter 1]

21 Aug 2024

1. The authors have now provided key research data, namely the total urine volume and bladder capacity. These data are very helpful to explore mechanisms underlying the voiding phenotype of T2DM model (OLETF) rats, but raised a serious concern about the authors’ interpretation/conclusion. Thus, increased voiding frequency could be simply due to polyuria but not bladder dysfunction. OLETF rats had x2.2 total urine volume and x1.8 voiding frequency at 36 weeks, x4.4 total urine volume and x3.9 voiding frequency at 48 weeks compared with control (LETO) rats. Tadalafil treatment reduced the total urine volume from x4.4 to x1.7 and the voiding frequency from x3.9 to x1.8 of control values. 

Morphological and functional changes in the bladder of OLETF rats could be a consequence of polyuria rather than the cause of increased voiding frequency. After all, the manuscript including the title should be largely rewritten. 

Response: Thank you for your comments. As the reviewer mentioned, the difference in the voiding frequency in LETO and OLETF rats might be mainly caused by the difference in total urine volume. Based on the results that tadalafil treatment decreased the urine volume in OLETF rats, it is suggested that tadalafil is effective for polyuria caused by T2DM. The changes in the total urine volume might be associated with the renal deficiency caused by T2DM. However, the mechanisms for them were not evaluated in this study. We are planning to make experiments in the future study. They were added in the Limitation section (Page 24, line 423) and the words “polyuria” and “renal” were added in the Conclusions section (Page 25 line438, 440, 443)

-Page 1, line 1. The title was changed from “PDE5 inhibitor potentially inhibits epithelial ATP release and recovers detrusor contractility in type 2 diabetic rats via increased bladder blood flow” inoto “PDE5 inhibitor potentially improves polyuria and bladder storage and voiding dysfunctions in type 2 diabetic rats”.

-Page 3, line 42. The conclusions of the abstract were changed as follows; “The T2DM rats had polyuria, increased ATP release induced by decreased bladder blood flow and impaired contractile function. PDE5 inhibition improved these changes and may prevent T2DM-associated urinary frequency and bladder storage and voiding dysfunctions.”

-Page 24, line 423. Several sentences were added.

-Page 25, line 438, 440 and 443. “Polyuria” and “renal” were added.

2. The results of cystometry lack statistical analysis, e.g., bladder capacity, number of non-voiding contractions or peak voiding pressure. 

Response: Thank you for your suggestion. As the results of cystometry, the bladder capacity, the bladder contraction pressure and the number of non-voiding contractions were shown. Moreover, several corrections were made in the methods of cystometry in conscious rats with an additional reference.

-Page 14, line 231. Sentences were changed as follows; “Cystometry in conscious rats was performed in the LETO and OLETF rats (n=2, each) at 36 weeks. The bladder capacity (0.7 ± 0.1 mL and 0.7 ± 0.2 mL, p=0.37), and the bladder contraction pressure (39.3 ± 10.7 cmH2O and 33.4 ± 10.3 cmH2O, p=0.16) were not significantly different between LETO and OLETF rats. Non-voiding contractions appeared only in OLETF rats and not appeared in LETO rats”

-Page 8, line 126. A sentence was changed as follows; “Cystometry procedures in conscious rats were performed as our previous reports(16)”.

-Page 8, line 133. A sentence “Cystometry was done with physiological saline at room temperature at 0.04 ml per minute. Bladder capacity and bladder contraction pressure were measured” was added.

3.The number of experiments (animals) should be provided along with p values throughout the text.

Response: Thank you for your suggestion. The number of animals used in each experiments were shown in the Results section.

-The number of rats were added as follows; Page 13, line 223 (n=6, each), Page 14, kine 239 (n=6, each), Page 15, line 253(n=3, each), Page 16, line 269 (n=3, each), Page 16, line 281 (n=3, each), Page 17, line 291 (n=3, each) and Page 18, line 303 (n=3, each).

4. Cystometric traces lack scale bars. In Fig1B, L-36 and O-36 traces appeared to be shown in different time scales (see the difference in the duration of voiding contractions). Please provide clearer images.

Response: Thank you for your suggestion. All the figures of cystometric traces were shown in a same scale. Scale bars were added in figure 1B. Then, the figures were made clearer.

-Figure 1 was changed.

5. The authors did not answer why atropine completely diminished KCL-induced contractions of DSM strips. This should not be happened if experiments were carried out properly (see my comment to the original submission).

Response: As the reviewer pointed out, the results of the effect of atropine for KCL-induced contraction is complicated. To prevent misunderstandings, the results were deleted in this manuscript. The figure 6C was deleted accordingly. 

-Figure 6 was changed.

6. Run out of animals is not a good excuse as both OLETF and LETO rats seem to be commercially available (http://www.hoshino-lab-animals.co.jp/English/products/OLETF_ENG..html). Yet, additional experiments are desirable but not mandatory in the present study. 

Response: Thank you for your comments. As the reviewer mentioned, the OLETF an LETO rats are commercially available. Additional experiments to elucidate the mechanisms of polyuria or the pathological changes in the bladder wall will be performed and make another paper.

Please check the accuracy of references (for example No.39). 

Response: The references (42 and 43) were corrected and the sentenced in Page 22, line 390 were also corrected.

---

## [Decision Letter · Decision Letter 2]

2 Sep 2024

PDE5 inhibitor potentially improves polyuria and bladder storage and voiding dysfunctions in type 2 diabetic rats

PONE-D-24-11736R2

Dear Dr. Terada,

We’re pleased to inform you that your manuscript has been judged scientifically suitable for publication and will be formally accepted for publication once it meets all outstanding technical requirements.

Kind regards,

Yung-Hsiang Chen, Ph.D.

Academic Editor

PLOS ONE

Additional Editor Comments (optional):

Congratulations on the acceptance of your manuscript, and thank you for your interest in submitting your work to PLOS ONE.

Reviewers' comments:

Reviewer's Responses to Questions

**Comments to the Author**

1. If the authors have adequately addressed your comments raised in a previous round of review and you feel that this manuscript is now acceptable for publication, you may indicate that here to bypass the “Comments to the Author” section, enter your conflict of interest statement in the “Confidential to Editor” section, and submit your "Accept" recommendation.

Reviewer #2: All comments have been addressed

Reviewer #3: All comments have been addressed

2. Is the manuscript technically sound, and do the data support the conclusions?

Reviewer #2: Partly

Reviewer #3: Yes

3. Has the statistical analysis been performed appropriately and rigorously? 

Reviewer #2: No

Reviewer #3: Yes

4. Have the authors made all data underlying the findings in their manuscript fully available?

Reviewer #2: Yes

Reviewer #3: Yes

5. Is the manuscript presented in an intelligible fashion and written in standard English?

Reviewer #2: Yes

Reviewer #3: Yes

6. Review Comments to the Author

Reviewer #2: (No Response)

Reviewer #3: The manuscript entitled “PDE5 inhibitor potentially improves polyuria and bladder storage and voiding dysfunctions in type 2 diabetic rats” has been well proofread.

7. PLOS authors have the option to publish the peer review history of their article (what does this mean?). If published, this will include your full peer review and any attached files.

Reviewer #2: No

Reviewer #3: No

---

## [Editor Report · Acceptance letter]

6 Sep 2024

PONE-D-24-11736R2 

PLOS ONE

Dear Dr. Terada, 

I'm pleased to inform you that your manuscript has been deemed suitable for publication in PLOS ONE. Congratulations! Your manuscript is now being handed over to our production team.

Kind regards, 

on behalf of

Dr. Yung-Hsiang Chen 

Academic Editor

PLOS ONE